

# L-band vegetation optical depth as an indicator of plant water potential in a temperate deciduous forest stand

Nataniel Holtzman[1], Leander D.L. Anderegg[2], Simon Kraatz[3], Alex Mavrovic[4], Oliver Sonnentag[5], Christoforos Pappas[5], Michael H. Cosh[6], Alexandre Langlois[7], Tarendra Lakhankar[8], Derek Tesser[8],

Nicholas Steiner[9], Andreas Colliander[10], Alexandre Roy[4], Alexandra G. Konings[1]

[1] Department of Earth System Science, Stanford University, Stanford, CA 94305, United States of America

[2] Department of Integrative Biology, University of California Berkeley, Berkeley, CA; Department of Ecology, Evolution and Marine Biology, University of California Santa Barbara, Santa Barbara, CA

[3] Department of Electrical and Computer Engineering, University of Massachusetts, Amherst, MA, USA

[4] Département des Sciences de l'Environnement, Université du Québec à Trois-Rivières (UQTR), Trois-Rivières, Québec, Canada

[5] Département de géographie, Université de Montréal, Montréal, Québec, H2V 2B8, Canada

[6] USDA ARS Hydrology and Remote Sensing Laboratory, Beltsville, MD, USA

[7] Département de Géomatique Appliquée, Université de Sherbrooke, Sherbrooke, Québec, Canada

[8] NOAA-CESSRST, The City College of New York, City University of New York, New York, NY, USA

[9] Department of Earth and Atmospheric Sciences, City College of New York, City University of New York, New York, NY, USA

[10] Jet Propulsion Laboratory, California Institute of Technology, Pasadena, CA, USA

*Correspondence to*: Nataniel Holtzman (nholtzma@stanford.edu)

**Abstract.** Vegetation optical depth (VOD) retrieved from microwave radiometry correlates with the total amount of water in vegetation, based on theoretical and empirical evidence. Because the total amount of water in vegetation varies with relative water content (as well as with biomass), this correlation further suggests a possible relationship between VOD and plant water potential, a quantity that drives plant hydraulic behavior. Previous studies have found evidence for that relationship on the scale of satellite pixels tens of kilometers across, but these comparisons suffer from significant scaling error. Here we used

small-scale remote sensing to test the link between remotely sensed VOD and plant water potential. We placed an L-band radiometer on a tower above the canopy looking down at red oak forest stand during the 2019 growing season in central Massachusetts, United States. We measured stem xylem and leaf water potentials of trees within the stand, and retrieved VOD with a single-channel algorithm based on continuous radiometer measurements and measured soil moisture. VOD exhibited a diurnal cycle similar to that of leaf and stem water potential, with a peak at approximately 5 AM. VOD was also positively

correlated with both the measured dielectric constant and water potentials of stem xylem over the growing season. The presence of moisture on the leaves did not affect the observed relationship between VOD and stem water potential. We used our observed





VOD-water potential relationship to estimate stand-level values for a radiative transfer parameter and a plant hydraulic parameter, which compared well with the published literature. Our findings support the use of VOD for plant hydraulic studies in temperate forests.


## 1 Introduction

To supply water for transpiration, plants transport water upwards from soil to leaf through their xylem tissue under negative pressure (tension). The rate of this transport process affects the water status of leaves – leaf water potential results from the balance of water lost to transpiration and water refilled through xylem. Through its effect on stomatal closure

(Venturas et al., 2017), leaf water potential in turn controls transpiration and photosynthesis rates. Accounting for plant hydraulics has been shown to improve models of stomatal conductance (Anderegg et al., 2017; Liu et al., 2020; Sabot et al., 2020; Wolf et al., 2016). This has motivated the recent inclusion of plant hydraulics in a number of land surface models (Christoffersen et al., 2016; Eller et al., 2020; Kennedy et al., 2019). Beyond influencing water and carbon fluxes, reductions in stem water potential (Adams et al., 2017) or water content (Rao et al., 2020) can cause drought-induced tree mortality, an

increased risk under rising temperatures (Williams et al., 2013) and evaporative demand (Novick et al., 2016). The dynamics of how water flows through vegetation can also affect fire risk (Nolan et al., 2020), crop yields (Konings et al., 2019), and phenology (Xu et al., 2016).

Spatiotemporally distributed data on plant water potential could therefore improve our global understanding of plant-water interactions, including aiding in the parametrization and testing of the latest generation of global land surface models.

However, current measurements of plant water potential are taken on individual plants, using psychrometers or pressure chambers. These methods are expensive and labor-intensive. Furthermore, they are difficult to scale up from the plant to the ecosystem level because in many stands, plants with very different hydraulic strategies and associated water potential dynamics grow together (Matheny et al., 2017; Skelton et al., 2015). If remote sensing data could provide signals related to plant water potential, it would naturally provide spatially aggregated and continuous data at scales relevant for land surface modelling

(parameterization and validation) and policy making (hot-spots of areas vulnerable to drought-stress). Passive microwave remote sensing is sensitive to the water content of vegetation through vegetation optical depth (VOD) and may therefore be a useful tool for monitoring ecosystem-scale plant water potential.

In grasslands and agricultural fields, VOD has been shown to be closely related to the total amount of water in vegetation (Jackson and Schmugge, 1991) based on a variety of campaigns with destructive measurements. Although

destructive measurements of water content are far more difficult in forests, electromagnetic theory suggests that this is also the case for forests (Ferrazzoli and Guerriero, 1996; Kurum et al., 2011). Furthermore, VOD at a range of electromagnetic frequencies has been found to scale with biomass in forests (Chaparro et al., 2019; Liu et al., 2015; Mialon et al., 2020) – a relationship that is formed through VOD's sensitivity to water content. Relative water content (which influences the canopy



water content per unit area observed by remote sensing) and water potential in vegetation are monotonically related, as has
been measured for countless species by ecophysiologists using so-called pressure-volume curves (Barnard et al., 2011; Bartlett
et al., 2012). VOD's sensitivity to vegetation water content therefore suggests it may also be sensitive to the water potential of
aboveground vegetation components, including leaves and stems. However, this has not yet been explicitly demonstrated.

Indirect evidence nevertheless suggests a relationship between VOD and leaf water potential. Konings and Gentine
(2017) showed that, if VOD is assumed to be linearly related to leaf water potential, it can be used to estimate ecosystem-scale
patterns of isohydricity around the world, displaying the expected global patterns. Momen et al. (2017) compared fluctuations
in satellite-base X-band VOD to leaf water potential measurements in three forest and woodland sites to *in situ* leaf water
potential measurements. After biomass changes were also accounted for through LAI, they were able to predict VOD with $R^2$
= 0.6-0.8. Zhang et al. (2019) extended this approach by estimating leaf water potential based on root-zone soil moisture
measurements in Oklahoma grasslands and using the assumption of pre-dawn water equilibrium between soil moisture and
leaf water potential. The resulting datasets were used in combination with NDVI to study the variations of X-band VOD ,
finding that while biomass changes (estimated through NDVI) were the dominant driver of VOD changes on time scales from
daily to seasonal, water potential did provide some additional utility in predicting VOD. Both the Momen et al. (2017) and
Zhang et al. (2019) studies suggest that leaf water potential may influence VOD, but the interpretation of both of those studies
is limited by the scale mismatch between water potential data (individual plants) and VOD data (pixels tens of km wide). In
this study, we aim to overcome the scale problem by using a microwave radiometer mounted on a tower, instead of satellite
data. The radiometer's 20 m by 25 m field of view is approximately five orders of magnitude smaller than the pixel size of
satellite-based VOD datasets. Furthermore, the field of view was dominated by a single tree species. At this scale, measuring
the water potentials of a few trees could give a good estimate of the average water potential of all vegetation within the
radiometer's view. Lastly, while microwave satellites typically make two overpasses per day for any given location, our
radiometer provided temporally continuous data, allowing us to capture the significant diurnal cycle in plant water potential.
We combined the tower-based radiometer with measurements of leaf and stem water potential, along with other environmental
data, to investigate three research questions:

a. How are VOD and plant water potential related at forest stand scale?
b. In a period of roughly constant biomass, how (if at all) does VOD change along with plant water potential
on time scales from hours to days?
c. What is the relative sensitivity of VOD to the water potential of woody stems, versus the water potential of
leaves?

We also note that the effects of electromagnetic observational frequency on the sensitivity and utility of VOD for
plant water stress studies remains uncertain. Past studies of VOD for plant water stress have mostly focused on X-band (i.e.
~10 GHz) VOD datasets from the Advanced Microwave Scanning Radiometer for EOS (AMSR-E) and Advanced Microwave
Scanning Radiometer-2 datasets, which allows creation of a relatively long data record (Du et al., 2017; Moesinger et al.,



2020). More recently, VOD datasets at L-band (~1.2 GHz) have also been derived from the European Space Agency (ESA)
Soil Moisture Ocean Salinity SMOS (Fernandez-Moran et al., 2017), and the National Aeronautics and Space Administration
(NASA) Soil Moisture Active Passive SMAP (Konings et al., 2017) satellites. The relatively longer wavelengths of L-band
observations reduce sensitivity to atmospheric humidity and increase penetration throughout the vegetation canopy. We
therefore focus on L-band observations in this study.

## 2 Methods

### 2.1 Field site

All data collection was conducted in 2019 near the hardwood walk-up tower in Harvard Forest (central Massachusetts,
USA, 42.54° N, 72.17 ° W). The site has a humid continental climate, with an average summer temperature of 17.9 °C during
June, July, and August 2019. The precipitation does not have a strong seasonality and totals to an annual value of about 110
cm (Waring et al., 1995). The site is a temperate deciduous forest dominated by red oak (*Quercus rubra*). Fig. 1 shows the
view from the top of the tower in early July, including the radiometer. The radiometer and certain other instruments were
installed at the site in late April and collected data until they were taken down in early December. An intensive field campaign
to collect leaf water potential data and to install additional instruments took place from July 9th to July 12th, with additional
shorter visits thereafter.

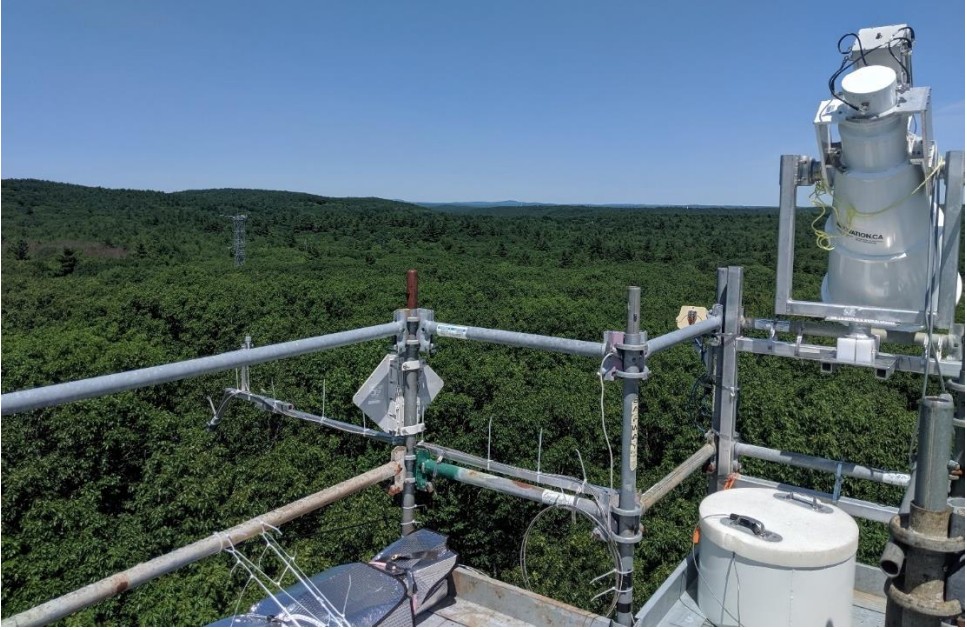

**Figure 1. Radiometer setup at Harvard Forest hardwood walk-up tower.**





## 2.2 Measurements

### 2.2.1 Microwave radiometer

As part of the SMAP Validation Experiment 2019-2021 (SMAPVEX19-21) campaign (Colliander et al., 2020), a downward-looking radiometer was installed at Harvard Forest. On April 28, 2019, a dual-polarization L-band (1.4 GHz) radiometer (Potter Horn PR-1475, Radiometrics Inc., Boulder, CO) was installed at 28 m above the ground surface on a double-scaffold tower, viewing the forest canopy from above at an oblique viewing angle. This radiometer has previously been used to study VOD and freeze-thaw state in a boreal forest (Roy et al., 2020). The PR-1475 radiometer has an antenna with a 30° half-power beamwidth (−3 dB) with an integration time of eight seconds. The hourly median brightness was used in further analysis to retrieve VOD. The antenna angle of incidence was adjusted manually with a hand-crank and a digital level. The radiometer was set to take continuous measurements of Brightness Temperature ($T_B$) above the canopy at an incidence angle from nadir of 40°. The footprint dimensions at 40° are 25 m long and 20 m wide. In addition, throughout the campaign, 15 calibrations were carried out using an ambient black body as a warm target and sky measurement as a cold target (5 K). Based on these calibrations, the radiometer accuracy at V-pol was approximately 2 K (Rowlandson et al., 2018). While both V-pol and H-pol brightness temperatures were measured, the H-pol data showed unexplained fluctuations throughout the campaign and were discarded from the analysis.

### 2.2.2 Plant physiological sensors

A variety of in situ soil and plant water sensors were also installed to better understand the drivers of the radiometric observations. However, due to logistical constraints, each of the instruments had a different observation period. The measurements are summarized in Table 1. Stem water potential was measured by PSY-1 stem psychrometers (ICT Instruments) installed on the main trunks of three trees at breast height. Two psychrometers were installed on July 9, 2019 and removed on July 12 of the same year. A third psychrometer was installed on July 10, and continued operating until July 17 when it ceased to collect realistic data, presumably due to extruded tree sap entering the sensor. This psychrometer was cleaned and reinstalled twice, both times collecting data for a few weeks before ceasing to collect data. In addition to the first operating period in July, the psychrometer operated from August 5 to August 27 and from September 5 to September 25. Stem xylem dielectric constant, electrical conductivity, and temperature were measured starting May 24 with two TEROS 12 soil moisture sensors drilled into the xylem at breast height (1.5 m). Further description of the use of soil moisture sensors in tree xylem can be found in Matheny et al. (2015). Note that the stem xylem dielectric measurements are performed at 70 MHz rather than the L-band (1.4 GHz) measurement of the radiometer. Five LWS leaf wetness sensors (METER Environment) were installed in the tower at canopy level on July 10. Each sensor recorded a binary reading (wet or dry) every 10 minutes. Hours where the majority of sensor-minutes were wet were considered wet for the purposes of our analysis; all other hours were considered dry.





### 2.2.3 Soil moisture and temperature

As a part of a the larger SMAPVEX19-21 experimental design, Stevens Water Hydra Probes were horizontally installed at 5 cm and 10 cm below the soil surface. An additional probe was installed vertically into the soil surface spanning 0 to 6 cm depth; this was the dataset used to in further analysis. Care was taken to not install through roots or substantial debris, but otherwise, these measurements are expected to capture representative soil moisture at the installation depth. Depths are approximate, as they sensing volume varies depending on soil moisture status and signal magnitude (where it is strongest close to the sensor and decreases away from a sensor. These sensors also measure soil temperature, with the vertically-installed sensor measuring a temperature at the soil surface. Air temperature was measured at a height of approximately 1 m by a Campbell Scientific 108 sensor within a radiation shield to protect the sensor from solar heating. There were three deployed stations within the radiometer footprint with occasional replacements for sensor or datalogger malfunction. For soil sensors and air temperature sensors, data was recorded every 30 minutes.


| Observation type | Model and manufacturer | Observation period (2019) |
|---|---|---|
| Vegetation optical depth at L-band | PR-1475 radiometer, Radiometrics Corporation | April 28 - October 17 |
| Stem xylem apparent dielectric constant at 70 MHz | TEROS 12, METER Environment | April 28 - October 17 |
| Leaf complex dielectric constant at L band | Custom time-domain reflectometer | July 9 - July 12 |
| Leaf wetness | LWS, METER Environment | July 10 - October 17 |
| Leaf water potential | M1000, PMS Instruments | July 9 - July 12 |
| Stem xylem water potential | PSY-1 psychrometer, ICT Instruments | July 9 - July 17, August 5 - August 7, September 5 - September 25 |
| Soil moisture and temperature | Hydraprobe, Stevens Water | April 28 - October 17 |
| Air temperature | Campbell Scientific 109 Air Temperature sensors | April 28 - October 17 |

**Table 1. Summary of data collected. Note that the stem xylem water potential sensor operated for three periods with gaps in between.**





### 2.2.4 Leaf measurements

165       The water potential of canopy red oak leaves was measured at 80 minute intervals between pre-dawn and sunset during a four day intensive observation period: the afternoon of July 9, all of daytime on July 10, the morning of July 11, and the morning of July 12. Leaf water potential was not measured in the afternoon of July 10 and late morning and afternoon of July 11, as it was raining then. To sample canopy leaves, we used grasping pole clippers to snip individual leaves that could be reached from the tower. Because of the radiometer viewing angle, these leaves did not fall in the radiometer footprint, but

they were within tens of meters of the footprint and had no obvious differences from the trees in the footprint. Thus, we assume the sampled trees are representative of trees in the radiometer footprint. By clipping leaves adjacent to the tower rather than using more complicated methods to collect leaf samples in the nearby footprint, we were able to quickly bag and measure each leaf less than 30 seconds after clipping, minimizing the possible error due to water loss in between the time of clipping and of measurement. At each collection time, three to five leaves each were cut from four trees, out of a set of five trees adjacent to

the tower. We did not collect leaves from all five trees at every collection time, because some trees had a limited number of leaves reachable from the tower. However, for any given collection time, leaves from at least four trees were gathered, and the trees from which leaves were gathered were alternated to reduce bias. Each leaf was wrapped in a moist paper towel to slow its dehydration right after clipping; we then immediately measured the leaf's water potential in a Scholander-style pressure chamber (PMS Instruments, Corvallis, OR). At several times of day following leaf water potential measurements, we used a

open-ended coaxial reflectometry probe (Mavrovic et al. 2018) to measure the L-band dielectric permittivity of a stack of the leaves that we collected (El-rayes and Ulaby, 1987). The leaves from different trees were inter-mingled so as not to bias the permittivity measurements towards a subset of the trees. These measurements were used to compare the sensitivity of VOD to both water potential and dielectric constant for both leaves and stem xylem.

      On the last day of the intensive observation period (July 12), three leaves and three 5 cm-long terminal branches were

collected pre-dawn and saved in closed plastic bags with moist paper towels. A pressure-volume curve relating water content to water potential was created for each of these samples by repeatedly measuring its mass and its water potential as it dried.

### 2.3 Vegetation optical depth (VOD) retrieval

      To retrieve VOD, we employed a single channel algorithm (SCA) using V-polarized L-band brightness temperature from the tower-based radiometer. Based on the zeroth order radiative transfer model commonly called the tau-omega model

(Mo et al, 1982; Ulaby and Long, 2014), brightness temperature at V- polarization can be written as follows:

$$T_{B,V} = (1 - r_V)\gamma T_s + \omega\gamma(1 + r_V\gamma)T_c , \qquad (1)$$

where $T_B$ is the V-polarized brightness temperature, $r$ is rough soil reflectivity in the same polarization, $\gamma$ is vegetation transmissivity, $\omega$ is vegetation single scattering albedo, $T_s$ is soil temperature, and $T_c$ is canopy temperature. Transmissivity is a function of VOD:

$$\gamma = \exp\left(-\frac{VOD}{\cos\theta}\right), \qquad (2)$$





where $\theta$ is the incidence angle of the sensor. Here, $\theta$ was fixed at $\theta = 40^{\circ}$, to match the observational conditions of the SMAP satellite. Single-channel algorithms for soil moisture retrieval commonly first assume a value of VOD and solve Eq. (2) for soil reflectivity, which is sensitive to soil moisture (Ulaby and Long, 2014). In this study we take the opposite approach, using soil moisture from in situ observations and solving for VOD. Once $r_V$ is known, Eq. (1) is exactly solvable for $\gamma$ (and thus for

VOD) if all other variables are known. As is common in satellite-based studies (Owe et al., 2001), we did not attempt to retrieve VOD during times when precipitation was occurring, to avoid VOD retrievals being influenced by water in the atmosphere as opposed to water in vegetation.

A common assumption in microwave radiometry is that the soil and canopy are in thermal equilibrium and their temperatures can be treated as equal, for early morning satellite overpasses (6:00 AM for both SMAP and SMOS) (O'Neill et

al., 2019). In this study, we use observations from all times of day, not just the early morning. Thus, we did not assume $T_s = T_c$, instead we used different sources of data for soil and canopy temperature. Soil temperature was measured at the soil surface. Although canopy biological temperature was not measured at the tower site, air temperature approximately 1 m height above ground level in the shade was measured. To provide confidence in its use as a proxy for the temperature of the canopy itself, we compared this air temperature dataset to thermal infrared measurements of canopy temperature at 16 m height from a station

less than 1 km away within Harvard Forest, part of the NEON network (National Ecological Observatory Network, 2020). The two temperature datasets were very similar (Supplemental Figure 1). Over the period from June to September (the approximate growing season), the Pearson's $r^2$ between the air temperature data and the NEON data was 0.98, and on average the air temperature was 0.57°C lower than the NEON temperature. By contrast, the Pearson's $r^2$ between the soil temperature and the NEON temperature was only 0.86, and on average the soil temperature was 1.0 °C lower than the NEON temperature. We

therefore used the in-situ air temperature as a proxy for canopy temperature at the site. Indeed, all our results are qualitatively unchanged when the NEON station temperature is used instead for canopy temperature. By contrast, our results do not hold when in-situ soil temperature is used as $T_c$ in the tau-omega model – counter to expectations, there is no significant mean diurnal cycle in the VOD time series retrieved with this approach  (although it was correlated with stem water potential on a multi-week time scale). The failure of the common $T_c = T_s$ assumption outside of predawn is understandable based on the large

divergence in afternoon temperatures between soil and canopy, as shown in Fig. 2. It is also in line with previous studies showing that afternoon soil-canopy temperature differences degrade the quality of the retrieved soil moisture (Lei et al., 2015; Parinussa et al., 2016).

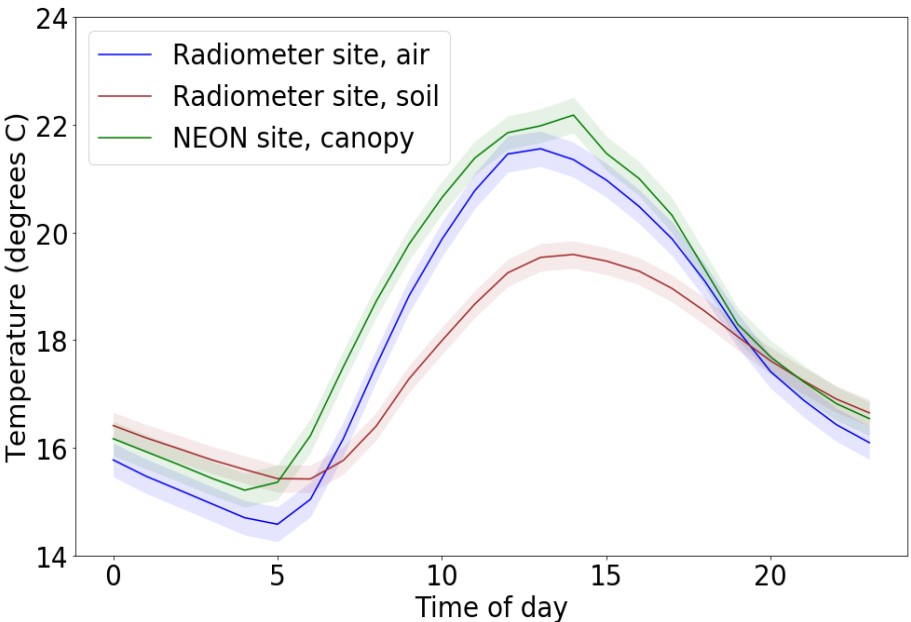

**Figure 2. Average diurnal cycles of temperature from 3 sources over June through September 2019. Shaded area is a range of 1 standard error. Air and soil temperatures each represent an average of two sensors. Time zone is local (Eastern Daylight Time)**

Values of several other parameters are needed to fully solve the tau-omega model. To obtain soil reflectivity values from soil moisture, we applied the Mironov dielectric mixing model to the in-situ soil moisture time series (Mironov et al., 2002) using a value of 9% clay content that McFarlane et al. (2013) measured at Harvard Forest. The scattering albedo ($\omega =$ 0.05) and the effect of soil surface roughness were parametrized as in the SMAP soil moisture product with parameters for temperate broadleaf forest. Using a different soil roughness correction shifts the retrieved VOD upward or downward but does not substantially change its trend or diurnal cycle. For example, doubling the RMS height in the roughness correction lowers the average VOD by 0.067 and results in a VOD that is correlated to the original VOD with $r^2 = 0.99$. Using a lower scattering albedo also shifts the retrieved VOD. For example, lowering the scattering albedo from 0.05 to 0.03 lowers the average VOD by 0.30 and results in a VOD that is correlated to the original VOD with $r^2 = 0.75$. By contrast, when a higher scattering albedo is used, the retrieval fails to converge for large periods of time (i.e. there are no solutions to the tau-omega equation for which $\gamma$ is a real number). For example, as the albedo is increased from 0.05 to 0.06 to 0.07, the fraction of hours where retrieval is successful decreases from 0.90 to 0.40 to 0.075.





**2.4 Interpretation of VOD-plant water potential relationship**

To aid in interpretation, we characterized the relationship between plant water potential and VOD using a simple multiplicative model, noting that vegetation water content (*VWC*) scales with both dry biomass (*AGB)* and the amount of water per unit biomass (relative water content, $RWC_B$). We use the same model as in Momen et al. (2017):

$$VOD = b * VWC = b * AGB * RWC_B \qquad (3)$$

Above, *b* is the slope of the relationship between VOD and total water content. In physiological studies, it is customary to define another type of relative water content: the water content of the plant divided by its maximum possible water content (i.e. the fully hydrated water content). We will call this quantity $RWC_H$. It is possible to convert between the two types of relative water content based on the average dry matter content of the plant (*DMC*, dry mass per total mass at full hydration):

$$RWC_B = \frac{1-DMC}{DMC} * RWC_H \qquad (4)$$

In this equation, the quantity *(1-DMC)/DMC* represents the ratio between water mass and dry mass for a fully hydrated plant.

While the relationship between $RWC_H$ and plant water potential is usually non-linear, especially at very low water potentials (Barnard et al., 2011; Bartlett et al., 2012), here we approximate the plant's pressure-volume curve over the typically-observed water potential range as a linear function:

$$RWC_H = 1 + \psi/\varepsilon \qquad (5)$$

In this equation, the maximum possible $RWC_H$ value of 1 is achieved when potential ($\psi$) is 0, and more negative values of potential produce a lower water content. The bulk modulus of elasticity $\varepsilon$ represents the change in water potential per change in $RWC_H$. Combining Equations (3), (4), and (5), VOD can be modeled as:

$$VOD = b * AGB * \frac{1-DMC}{DMC} * (1 + \psi/\varepsilon) \qquad (6)$$

When biomass is constant, Eq. (6) takes the form of a linear relationship between plant water potential and VOD. In this study, we measured changes in VOD and plant water potential during a period of several days in midsummer, during which biomass was assumed to be constant. A linear function was fitted to the observed relationship between VOD and leaf water potential, and values of the $\varepsilon$ and *b* parameters were calculated from the slope and intercept of that function.

**3 Results**

**3.1 Temporal dynamics of VOD and plant water status**

The retrieved VOD time series ranges from 0.18 to 2.09, with a mean of 1.00, a 25[th] percentile of 0.87, and a 75[th] percentile of 1.14 (Supplemental Figure 2). These are realistic values for a dense forest (Konings et al., 2017). There is a declining trend of VOD over the course of the summer from June through September (slope = -0.10 ± 0.0033 /month), which may correspond to drying conditions; soil moisture also showed a decreasing trend over the same period (slope = -0.053 ±





0.00086 cm³/cm³/month). During the week when leaf water potential was observed (Fig. 3), a multi-day decreasing trend is found in VOD, stem xylem dielectric, and stem xylem water potential from July 11 through July 14, which levels off or slightly increases over the next three days. Those three variables, as well as leaf water potential, all show a diurnal oscillation. They are lowest around midday and afternoon, and highest between midnight and pre-dawn hours. This daily cycle in water potential and stem xylem dielectric has been observed extensively in prior studies, and represents the signature of plant water usage

(Klepper, 1968; Matheny et al., 2017). During daytime the plant loses water to transpiration, and during night it refills its water by drawing on soil moisture.

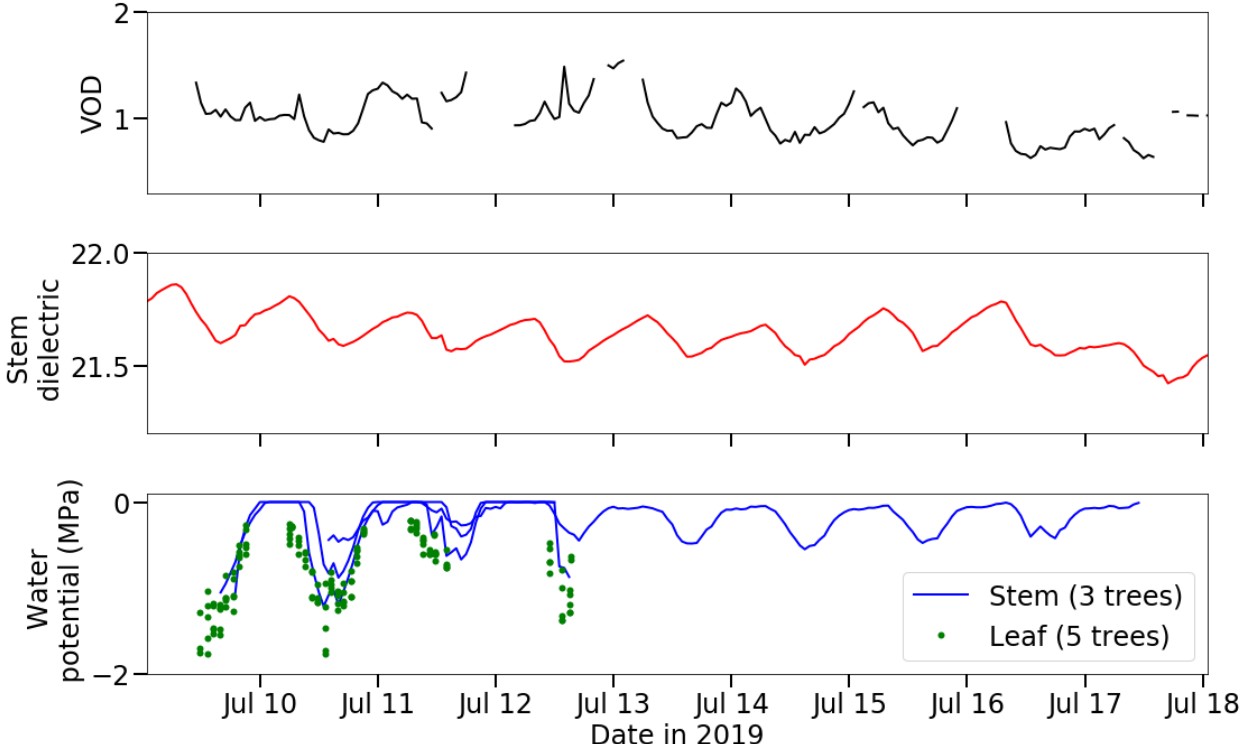

**Figure 3. VOD, stem xylem dielectric constant at 70 MHz, and plant water potential during the intensive observation period.**


The similarity between the diurnal patterns of water potential and VOD is even more apparent when the entire VOD record is composited into an average daily cycle (Fig. 4). In this view, leaf water potential starts decreasing approximately 3 hours before stem xylem water potential does. This lag has been seen in models and field studies (Zweifel et al., 2001), and is due to the leaves being exposed to the sun and drying out faster than the signal of decreasing water potential propagates down

to the lower trunk. The diurnal course of VOD starts decreasing early in the morning, with the magnitude, start of the decline and daily minimum time all more similar to leaf potential than to stem xylem potential. The implications of this difference are





discussed further in Section 4.3. The average diurnal cycle of VOD over only July 9 through July 17 is not as smooth as that in Fig. 4, but it has the same qualitative features (Supplemental Figure 3).

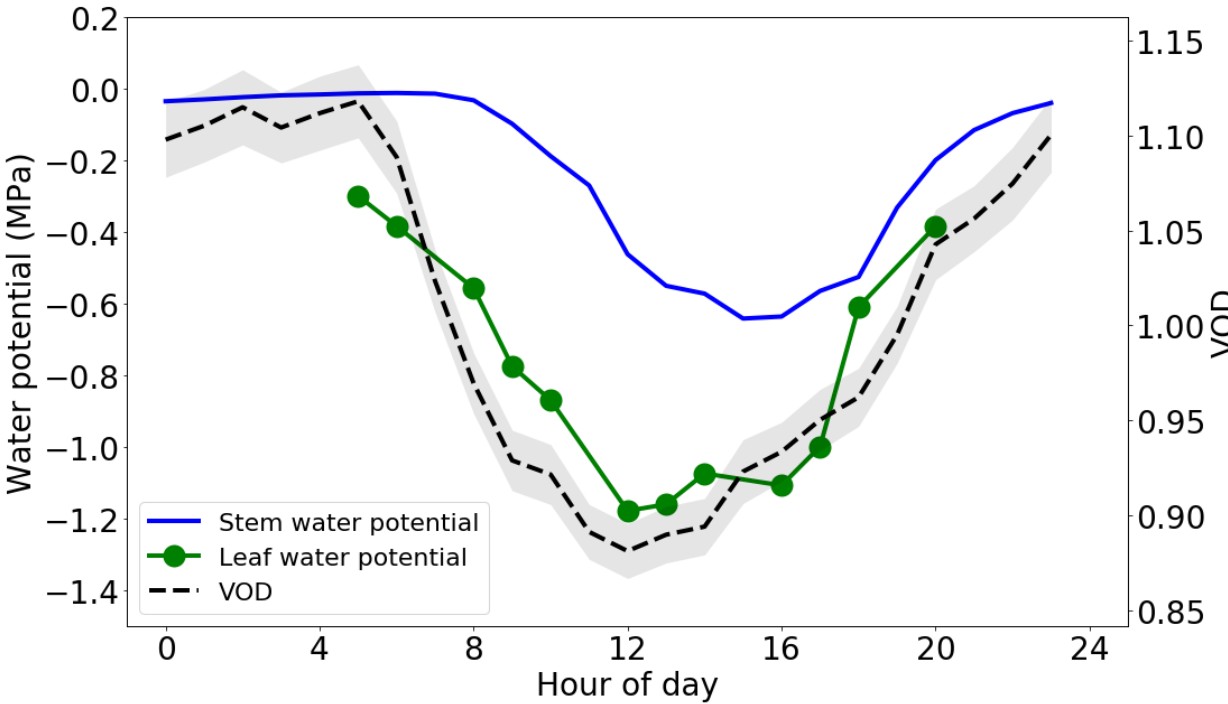

**Figure 4. Average diurnal cycles of VOD and plant water potential. Shaded area is a range of 1 standard error for VOD. Note that the three variables are each averaged over different periods because of differing record length (April-October for VOD, July 9-12 for leaves, and July 9-17 for stem xylem).**

### 3.2 Leaf and stem influence on VOD

When simultaneous measurements of VOD, leaf water potential, and stem xylem water potential are all compared, VOD is strongly positively correlated with both average stem and average leaf water potential (Fig. 5). However, the VOD-stem water potential relationship breaks down at very wet values of stem xylem water potential in the early morning hours, possibly due to lack of sensitivity of the psychrometer in this regime. It is not clear from this analysis which part of the plant influences VOD more. When using linear regression to predict VOD from a weighted average of leaf and stem potential, the leaf potential has approximately 1.5 times the weight of stem potential ($VOD = 1.15 + 0.18\psi_{leaf} + 0.12\psi_{stem}$ , $R^2 = 0.66$, p $< 0.0001$). However, the results of this regression should be taken with caution for two reasons. First, collinearity between leaf and stem xylem potential means the weights have very large standard errors ($0.18 \pm 0.08$ and $0.12 \pm 0.07$, respectively). Second, we measured more trees for leaf potential (n = 5) than for stem xylem potential (n = 2) during the time period where leaf and stem measurements overlapped, so we would expect less noise in the leaf measurements once they are averaged.

 

The parameters $\varepsilon$ and $b$ in Eq. (6) can be estimated using the measured relationship between leaf water potential and
VOD shown in Fig. 5a. To do so, we assumed a typical above-ground biomass value for Harvard Forest of 12.5 kg/m² (Munger
and Wofsy, 2020). Furthermore, we assumed that the tree-scale dry matter content falls in the range of 0.37 to 0.57 that Palacio
et al. (2008) observed in oak branches in Spain. Depending on the unknown DMC at our site, we estimate a possible range of
0.055 to 0.13 for $b$, the slope between VOD and the total water content *VWC*, and a value of 4.1 MPa for $\varepsilon$, the modulus of
elasticity. Results were similar when stem water potential was used instead of leaf water potential in this procedure, yielding
estimates of 0.050 to 0.11 for $b$ and 4.4 MPa for $\varepsilon$.

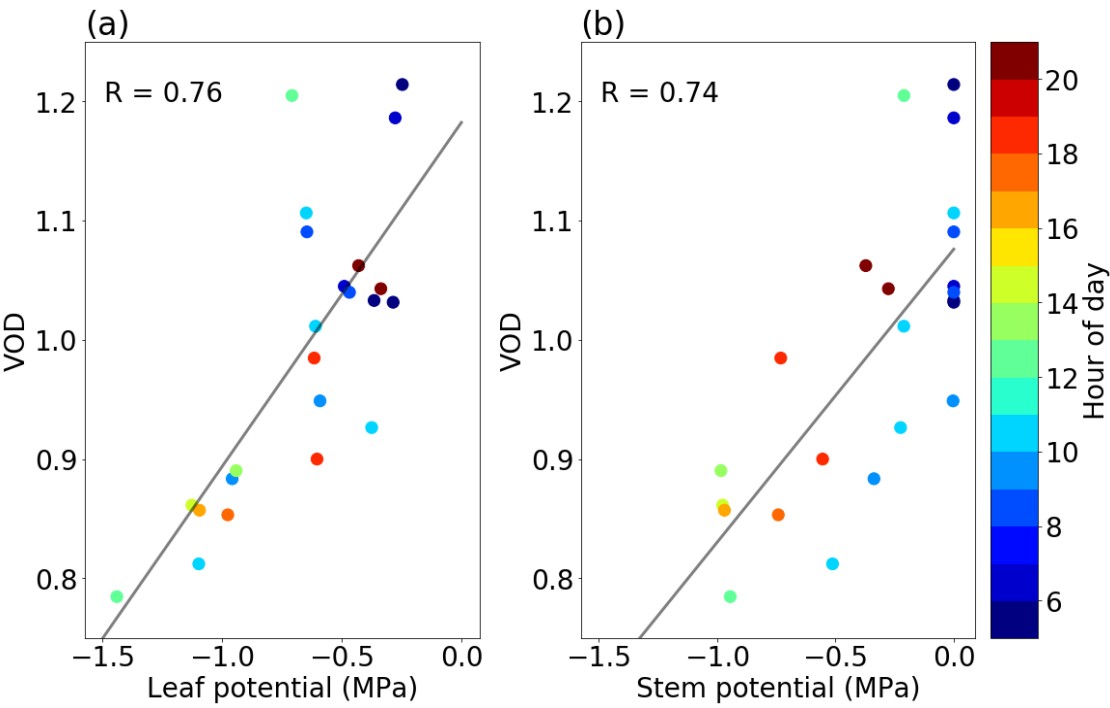

**Figure 5. Scatter plots and regression lines of VOD compared to leaf and stem xylem potential, averaged over all
samples at each time point, during July 9-12, 2019.**

      Based on electromagnetic theory, the L-band dielectric constant (rather than water content or potential) is the physical
variable that should directly control L-band VOD. Interestingly, as shown in Fig. 6, leaf water potential can actually predict
VOD (Fig. 6c) slightly better than direct measurements of L-band leaf dielectric constant can (Fig. 6b). This finding may be
due to differing noise levels in the measurement systems we used for potential compared to dielectric constant. The VOD-leaf
potential correlation in Fig. 6 is different from that in Fig. 5, because the former is limited to leaf water potential observations
that coincided with a leaf dielectric constant observation. Over several months, VOD is positively correlated with stem xylem





water potential, as well as with stem dielectric constant at 70 MHz (Fig. 7). This finding suggests that the potential-VOD

relationship holds over the entire growing season.

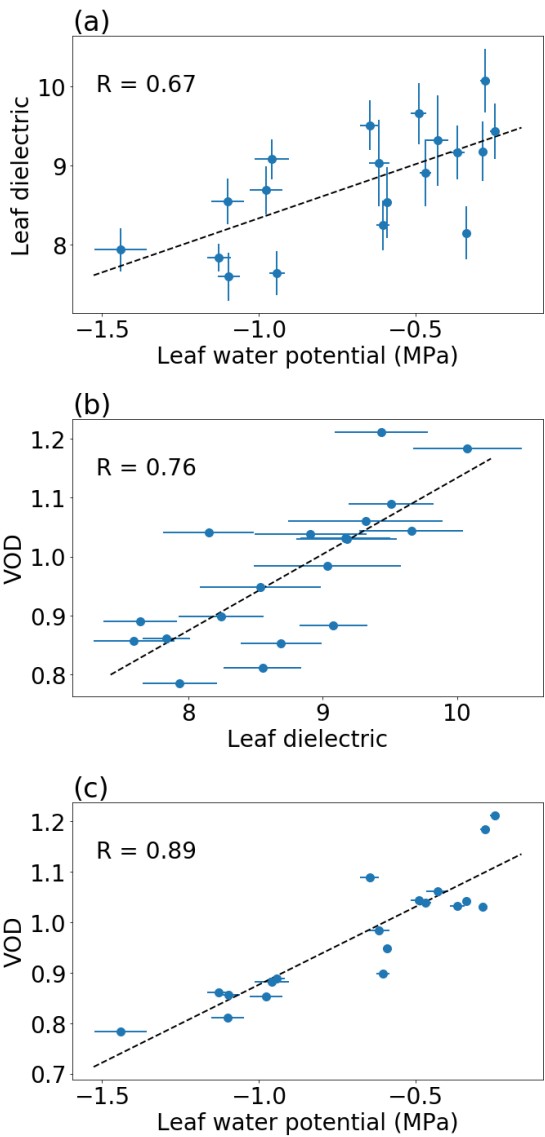

**Figure 6. Scatter plots of leaf water potential, leaf dielectric constant at L band (real part), and VOD. Leaf**
**measurements were taken July 9-12. For leaf water potential, each point represents a mean of several single-leaf**
**measurements per tree from five trees. For leaf dielectric, each point represents a mean of measurements from five**
**trees, with multiple vertically stacked leaves per tree contributing to each measurement. Error bars represent 1**
**standard error. The dashed line represents a linear regression.**




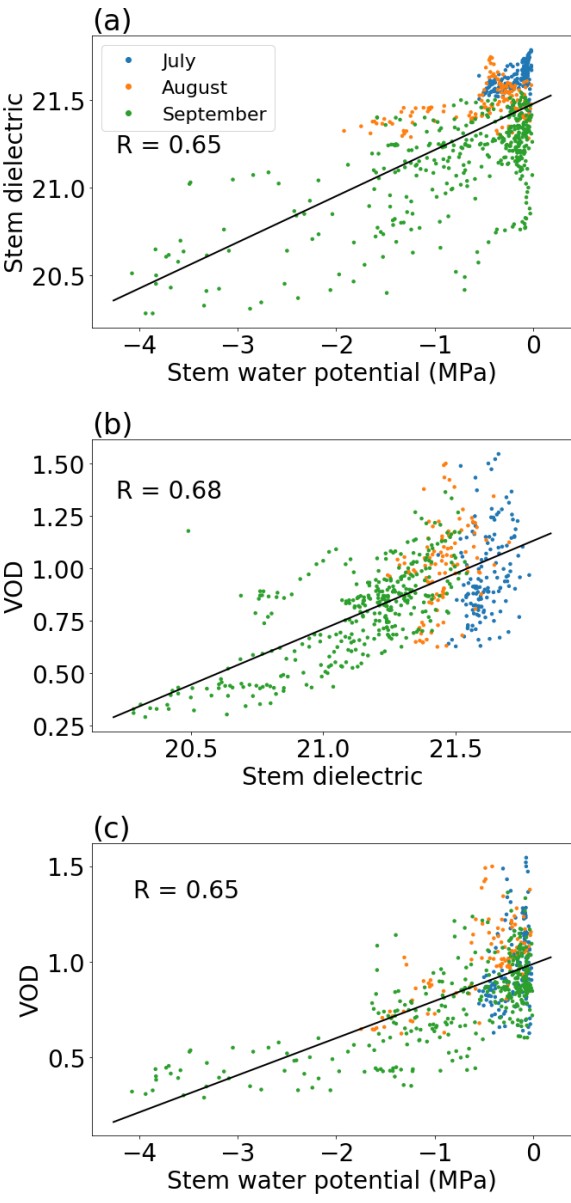

**Figure 7. Scatter plots of stem xylem water potential, stem xylem apparent dielectric permittivity at 70 MHz, and VOD with linear fits. Stem measurements are from a single tree. The three months labelled by color correspond to three successive installations of the stem psychrometer used to measure stem water potential.**






### 3.4 Canopy interception fails to influence VOD

To assess whether VOD was affected by water on the surface of vegetation, we compared the relationship between VOD and stem xylem dielectric permittivity during times when leaf wetness sensors showed the canopy was wet, versus times when the canopy was dry (Fig. 8). Stem dielectric was used in place of leaf or stem water potential in this analysis, because

the short length of stem and leaf water potential data sets meant they contained very few times in which the canopy was wet. We fit a linear model to predict VOD from two variables: stem apparent dielectric constant, and a binary variable that was 1 when the canopy was wet and 0 otherwise. The coefficient of the binary wetness variable was not significantly different from zero ($p > 0.25$), neither between midnight and 9 AM (when canopy wetness is most likely dew) nor between 10 AM and 11 PM (when canopy wetness is most likely intercepted rainfall).


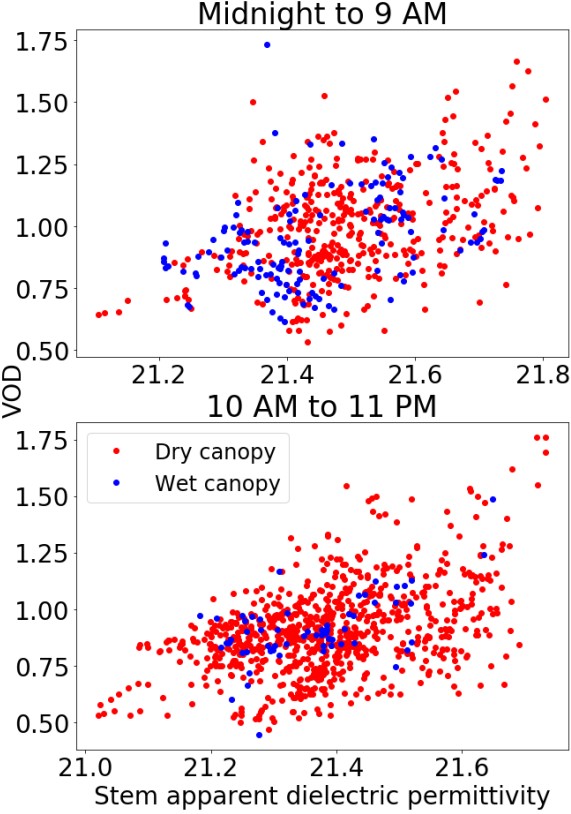

**Figure 8. Scatter plots of VOD vs stem permittivity, colored by canopy wetness. Measurements after September 17 were excluded from this analysis because of a large change in the shape of the VOD-dielectric relationship at that time, as discussed in section 4.1.**






## 4. Discussion and conclusions

### 4.1. Strong and approximately linear relationship between VOD and plant water potential

Our study demonstrates that VOD and plant water potential are closely related in a homogeneous temperate broadleaf forest during a single growing season. In theory, the existence of the VOD-water potential relationship should be applicable
to other land cover types as well, but further research is needed to confirm its applicability in a wide range of land cover types. Of the steps from water potential to dielectric constant to VOD shown in Fig. 6 and 7, there is relatively little nonlinearity, and none is present for leaves. Theoretically, the relationship between leaf water potential and RWC can be distinctly non-linear, particularly at very negative water potentials below the turgor loss point of leaves and for leaves with a large amounts of apoplastic water outside of cells (Bartlett et al., 2012). Indeed, the pressure-volume curves we measured for leaves and small
branches collected from Harvard Forest in July were fairly linear for water potentials above the turgor loss point, which was approximately -1.7 MPa (Supplemental Figure 4). Plant communities elsewhere may have more nonlinearity in their aggregate potential-VOD relationship, based on their exact pressure-volume curves and water content-dielectric relationships. Such nonlinearity might also occur at the Harvard Forest site under conditions we did not observe in 2019, such as extreme drought. More research is needed to understand the functional forms of the step-by-step links between water potential, water content,
plant dielectric, and VOD in various plant communities. Nevertheless, if confirmed elsewhere, our results suggest that future studies using VOD to understand plant water potential dynamics may be able to assume a simple relationship between plant water potential and VOD during times of constant biomass, which would greatly simplify their interpretation.

Unlike for the leaf data (Fig. 6), there is increased noise in the stem xylem potential-VOD relationship compared to the stem xylem dielectric-VOD relationship (Fig. 7), although the difference in the amount of explained variability is small.
Possible explanations for the higher noise level include the fact that the stem dielectric sensor stayed installed for the whole observation period, while the psychrometer was removed and reinstalled several times for cleaning, which could have placed it in different patches of xylem. We cannot rule out biases introduced by reinstallation being partially responsible for the extremely low water potentials the psychrometer detected in September (as low as -4 MPa). The lowest water potentials measured by the psychrometer co-occur with the lowest values of stem xylem dielectric, VOD, and soil moisture, so they do
appear to represent a particularly dry period. The slope of the VOD/stem water potential relationship is significantly smaller for water potentials lower than -2 MPa (which only occurred during the driest part of September) than for water potentials greater than -2 MPa (Fig. 7c). This change in slope may be an artifact of the psychrometer installation or calibration, or it may represent inherent nonlinearity in the pressure-volume curve of the oaks in our field site. The pressure-volume curves we measured for leaves and branches only contain water potentials greater than -2.5 MPa, so they do not provide information on
hydraulic behavior at very low water potentials (Supplemental Figure 4). However, a nonlinear pressure-volume curve in which the slope of the water content/water potential relationship $(1/\varepsilon)$ decreases as water potential becomes more negative has been found in a hydraulic model of oaks (Mirfenderesgi et al., 2016). A water content/water potential relationship with that



convex shape could lead to the same convex shape in the VOD-water potential relationship we observed at Harvard Forest (Fig 7c).

The relationship between stem xylem water potential and dielectric constant loses sensitivity at high values, as also seen in Fig. 7. These high values mostly occur at night-time, but could also occur under very wet conditions. This could be attributable to either a loss of sensitivity of water content to water potential at near-zero stem water potential values, and/or a loss of sensitivity of the dielectric constant to water content at high water content values, as has previously been observed in some species (Razafindratsima et al., 2017). As shown by the different colors of points in Fig. 7a, the y-intercept of the

relationship between stem dielectric and xylem water potential changes over the growing season, indicating a drift in dielectric constant that is not representative of water potential changes. Previous studies have found that stem xylem dielectric is sensitive to temperature and sap chemistry (in addition to xylem water content) and may vary significantly based on the sensor's location within a tree (McDonald et al., 2002). Thus, we may not expect all observed changes in stem dielectric to be reflected in xylem water potential. A drift over time in the relationship between stem dielectric and VOD is also observable in Fig. 7b. It is

possible that this drift represents conditions specific to the individual tree in which the dielectric sensor was installed, which may average away when scaling up to the scale of the radiometer footprint. Such a drift is not present in the stem potential-VOD relationship, despite stem potential being measured on the same single tree as xylem dielectric. A possible explanation for this difference is that the hydraulic behavior may have been relatively uniform among trees in the stand, while the other conditions that influence stem dielectric (e.g. sap chemistry) varied more from tree to tree.

## 4.2. Interpreting the VOD-water potential relationship in physiological terms


From the relationship we observed between VOD and leaf water potential, as well as ancillary data on biomass and tree dry matter content, we estimated $b$, the slope between VOD and total water content $VWC$, to lie in the range of 0.05 to 0.13. We can compare this estimate to values assembled from several studies of agricultural fields using radiometry in H polarization using destructive measurements of vegetation water content (Van de Griend and Wigneron, 2004). For L band

measurements, the $b$ values in Van de Griend and Wigneron (2004) ranged from 0.05 to 0.182, containing the range we calculate in this paper. However, it should be noted that relatively little is known about the b-factor in forests, where destructive comparisons of VOD and VWC have not been performed. Furthermore, the comparison is made more uncertain because of the difference in polarization between prior measurements at H-pol and our observations at V-pol. Nevertheless, the fact that our estimate of $b$ falls within the range of prior observations, despite deriving from a different land cover type and different

methodology, lends confidence to our analysis.

We estimated a value of 4.1 MPa for $\varepsilon$, the vegetation bulk modulus of elasticity that relates $RWC_H$ and $\psi$ (or 4.4 MPa based on stem xylem instead of leaf water potential data). We can also calculate this parameter directly from the pressure-volume curves we measured on individual leaves and twigs collected from Harvard Forest (Supplemental Figure 4). The pressure-volume analysis gives elastic moduli of 16 and 18 MPa for branches and leaves, respectively – much larger than the

value calculated based on VOD. Several previous studies that analyzed pressure-volume curves for oaks also found relatively





larger elastic moduli compared to our VOD-based value. Bahari et al. (1985) found elastic moduli ranging from 6.6 to 8.8 MPa in red oak leaves from a temperate forest in Missouri. Corcuera et al. (2002) found a range of approximately 10 to 20 MPa for elastic moduli in leaf-bearing shoots from 11 species of temperate-climate oaks.

An elastic modulus calculated from VOD represents, in a sense, the slope of an effective pressure-volume curve across the entire stand, aggregating the roles of leaves, branches, and trunks in proportion to their contribution to VOD. To our knowledge, no study has simultaneously measured the elastic moduli of these three vegetation components in oaks. In the palm tree *Sabal palmetto*, Holbrook and Sinclair (1992) measured a modulus of elasticity that was 346 times smaller for stem parenchyma than for leaves. If their observation of a smaller modulus of elasticity for trunks than for leaves holds qualitatively for oaks, then the fact that our VOD-derived elastic modulus is lower than leaf-derived values is to be expected, as the VOD-

derived estimate includes a contribution from trunks that would make the aggregate value lower than the leaf value. Model-based studies may provide further details on how effective pressure-volume curves scale from tissue to tree to stand. For a stand in Michigan containing a mixture of species, 96% of which were oaks, Mirfenderesgi et al. (2016) used a plant hydraulic model calibrated with sap flow measurements to infer an aggregate stem pressure-volume curve with an elastic modulus of approximately 5.0 MPa, which is closer to our VOD-based value than the values based on leaf or shoot pressure-volume curves

are.

### 4.3. Contributions of vegetation components to VOD: upper canopy vs. tree trunks

    Although our leaf observations only cover a few days in time, they coincide with a range of the 11[th] through the 89[th] percentile of VOD values relative to the entire April-October dataset. This wide range suggests that our results are adequate to characterize the VOD-leaf potential relationship throughout the growing season at our site. However, it should be noted

that LAI was relatively constant during this period, so that this study is unable to determine the relative roles of LAI and water potential in VOD variations during periods when LAI varies. The contributions of LAI (and biomass more generally) to variations of VOD can be substantial (Momen et al., 2017; Zhang et al., 2019).

    Our results also do not provide much insight whether L-band VOD is more sensitive to leafy or woody biomass over time scales longer than a day, because we were not able to measure leaf water potential or leaf dielectric at multiple points in

the growing season as we did for the stems. However, leaf and stem water potentials are highly correlated, and indeed mechanistically linked, on an inter-day time scale (Lambers et al., 2008). Thus, the positive relationship between VOD and stem xylem water potential across the growing season (Fig. 7) should hold qualitatively for leaves too. That is, VOD and leaf water potential are expected to be correlated across the entire growing season.

    Diurnal variations provide further information about the relative sensitivity of VOD to different canopy components.

Between 5 AM and 8 AM, stem xylem potential stays high while VOD and leaf potential begin to decrease, as shown in Figs. 5 and 6. As shown from the individual scatter points in Fig. 5b, VOD is much less closely related to stem water potential during the morning than it is at mid-day. In addition, the lowest point of the day for stem potential is around 3 PM, while for





both leaf potential and VOD it is between 12 and 1 PM. This suggests that VOD may be more influenced by the water status of leaves (and upper branches) than that of trunks. This is consistent with the notion that grey body emission is attenuated by
vegetation, such that overall observations are less sensitive to vegetation layers closer to the ground. It is also consistent with airborne observations showing L-band brightness temperatures differed significantly between poplar trees with foliage and the same defoliated trees later in the season (Santi et al., 2009) and with model analyses suggesting that at L-band, the majority of a canopy's contribution to observed brightness temperatures  is due to branches (Ferrazzoli and Guerriero, 1996; Paloscia et al., 2000). While we did not measure branch water potential, we expect branches in the canopy to have a potential closer to
that of canopy leaves than that of trunks at breast height.

Nevertheless, the relatively larger sensitive of VOD to leaf water potential than to stem water potential is notable because most of a tree's mass is in its trunk. Based on data in the literature for leaf mass, branch mass, and trunk mass of oak trees, we estimated that approximately that only 21% of an oak tree's water is expected to be in its branches and leaves with 79% in its trunk (see Supplemental Information for calculation), illustrating the dominant effect of attenuation on the sources
of the VOD signal even at L-band.

### 4.4. Implications for remote sensing of VOD

Our study highlights the importance of considering differences in canopy and soil temperature (Fig. 2) when retrieving VOD during the afternoon in densely vegetated areas. To investigate the information lost when neglecting temperature gradients (as is commonly done), we conducted an alternative retrieval of VOD assuming the canopy temperature equals the
soil temperature throughout the day. VOD retrieved with this method did not show significant diurnal variation. Thus, if diurnal or afternoon data is of interest, ecohydrological studies of VOD may benefit from VOD retrievals that account for soil and canopy temperature differences.

The presence of dew in the canopy did not alter VOD from what would be expected given the stem xylem water status (Fig. 8). This is consistent with Escorihuela et al. (2009), who did not find dew to have an effect on observed brightness
temperatures at L-band over a grassland, and Rowlandson et al (2012), who did not see a consistent effect of dew on L-band observations over a corn field (although an intermittent effect could not be ruled out). However, rainfall interception in grasslands can moderately increase L-band VOD (Saleh et al., 2006). Overall, the effect of leaf surface wetness on L-band VOD may depend strongly on the canopy type and droplet amount. Forests may be less sensitive to leaf surface water because the leaf surface water represents a small amount of the total water volume, or because the different shape of the leaves collect
water differently, leading to more or less water running off the leaves or differences in typical droplet sizes. It should be noted, however, that leaf surface water has been found to significantly influence observed X-band brightness temperatures at a tropical forest in Panama (Schneebeli et al., 2011). More research is needed to better understand how VOD sensitivity varies between water internal and external to the canopy. Nevertheless, our findings are encouraging for the use of early-morning VOD, and for approaches that compare VOD across different times of day based on the notion that pre-dawn leaf water potential is in
equilibrium with root zone soil water potential (Konings and Gentine, 2017). Additionally, many studies using VOD for studying plant water stress response filter out VOD shortly after rainfall to avoid noise from rainfall interception (Konings et al., 2017; Konings and Gentine, 2017), which in turn may bias studies towards periods in dry seasons where fewer data are filtered out. If the lack of sensitivity to rainfall interception we observed can be confirmed, this would reduce unnecessary data filtering.

490       More generally, the observed relationships between VOD and plant hydraulic quantities in this study reflect only a single stand and do not account for significant changes in biomass. To fully mature the use of passive microwave radiometry for studies of plant water use, we recommend additional validation field studies measuring leaf or stem water potential, as well as further study of differences in water potential – water content relationships between species and ecosystems, as well as the electromagnetic effect of stand type and vegetation geometry on the sensitivity of VOD at different frequencies to water

potential in different tree components. Improved understanding of these issues will enable new applications of passive microwave remote sensing.

**Author contributions**

AGK, NMH, LA, AR and AC designed the experiment. NH, LA, SK, AM, OS, CP, MC, AL, TL, DT, NS, AR, and AGK installed sensors and collected data. NH, LA, and AGK interpreted the data, with contributions from all authors. NH and AGK

wrote the first draft of the manuscripts, and all authors assisted in editing the manuscript.

**Data availability**

The authors have submitted relevant data to the National Snow and Ice Data Center (NSIDC). The final version of the paper will include a publicly accessible link to the data with a DOI.

**Competing interests**

The authors declare they have no competing interests.

**Acknowledgements**

NMH received funding from the NASA "Future Investigators in NASA Earth and Space Science and Technology" (FINESST) program and a Stanford McGee/Levorsen research grant. AGK and NMH were also supported by NASA Terrestrial Ecology award 80NSSC18K0715 through the New Investigator Program, by the NASA Carbon Cycle Science program, by NOAA

grant NA17OAR4310127, and by the Stanford Woods Institute for the Environment. AR, AM, OS and AL were supported by the Canadian Space Agency, the Natural Sciences and Engineering Research Council of Canada and the Canadian Foundation





for Innovation. AC made his contribution to this work at Jet Propulsion Laboratory, California Institute of Technology under a contract with NASA. We are grateful to Mark VanScoy and Audrey Plotkin-Barker for logistical support at Harvard Forest.

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
