# Peer review of "L-band vegetation optical depth as an indicator of plant water potential in a temperate deciduous forest stand"

_Biogeosciences, 2020_

## Referee Comment (RC1) · Anonymous Referee #1 · 12 Nov 2020

General comments:

The manuscript presents the use of tower-based radiometer data in L-band to test the link between estimated VOD from one polarization and plant water potential over the red oak forest. The manuscript reports on an experiment carried out during the 2019 growing season in central Massachusetts, United States. The manuscript is well written and contains worthwhile material that brings up the diurnal variation in VOD and its relationship with the diurnal plant water potential cycle. These results are credible and could further the understanding of the crop water dynamics. However, the intensive fieldwork to collect leaf water potential data has been done for 4 days in July which

was quite short in time and it was not enough to make a firm conclusion. Although short filed work experiments have been performed, the presented results can be used in future analysis. Therefore, I think this work can be accepted for publication.

Specifics comments:

1- Please add the Precipitation data and dew data in the manuscripts or supplemental information.

2- It would be nice if you compare VOD results from SMAP satellite with your data for the study area to illustrate the effect of scale on VOD estimation and especially shows how much VOD estimation would be different between your data and SMAP on the evening overpasses.

3- Although some of the previous studies have indicated that the presence of dew cannot affect the VOD, some studies presented its effect on VOD estimation. It would be nice if you can make a plot and add it in the supplementary in which shows the averaged VOD value at each hour from midnight to early morning for example from 12 to 6 for two conditions of the wet and dry canopy. It will help to better investigate the effect of dew on VOD estimation.

4- In figure 3, the temporal variation in VOD is different for each day. It would be nice if you could discuss the reason for the different daily trends. Also, the decreasing trend that you mention from July 11 through July 14 is not visible from the graph and it would be useful if you can fit a line to show that trend in the supplementary material.

5- In figure 7, you fit a line to the scatter plots and consider a linear relationship. By checking data of 3 months, we can see that the data in September shows some linear trend as the fitted line shows but the data of June and July are not fitted to that line. So, I think it would be better for this plot that you will use the Spearman rank correlation method and also shows the correlation for each month separately.

---

## Referee Comment (RC2) · Anonymous Referee #2 · 17 Nov 2020

In this manuscript, the relationship between VOD and plant water potential is analysed using an L-band radiometer and in situ measurements of stem xylem and leaf water potential and dielectric constants. In addition to the relationship in general, the authors investigate diurnal changes of VOD and the sensitivity of VOD to the stem and leaf water potential, respectively. The authors provide a comprehensive overview of basic plant hydraulics, the applied VOD retrieval and all conducted measurements. A weakness of the study is the limited number of in situ samples especially from leaves, but the authors are aware of that and describe the associated uncertainties. Despite the low number of samples, the findings presented in the manuscript are interesting and contribute to a better understand the variables which affect VOD over temperate forest.

[Figure]

Some detailed comments:

**266f: In the second half of September, VOD, stem dielectric constant and potential drop significantly. I saw that you refer to this later but consider mentioning it already here.**

Figure 3c: Do you have an idea why one of the stem water potential curves is very close to leaf water potential on July 10?

Figure 4: The VOD curve presents the average from April-October. You show in the supplement Fig. 3 that the VOD from July 9-12 does not differ much from the April-October average apart from the absolute values. Have you considered showing both figures in the main part of the manuscript, or adding the July-VOD to Fig. 4? In my opinion this would add information, as April-October is almost the entire growing season, whereas in mid-July not many LAI/biomass-related dynamics occur in temperate forest.

**296f/Figure 5a: The VOD-leaf water potential relationship also seems to break down, but during morning and evening hours and at a leaf potential around -0.5 MPa. Can you elaborate on this?**

Figure 5/6: You obtain a much better relationship in Fig. 6c than in 5a, just by leaving out the measurements during which you did not measure the leaf dielectric constant (R=89 vs. R=76, no "break down"/vertical linear relationship at -0.5 MPa in Fig. 6c). Can you explain if there is any reason for this? When did you measure the leaf dielectric constant, when not - just randomly?

**324f/Figure 7: I agree that there is a linear relationship over the entire growing season. But when looking at the individual months, there is a clear difference in the slope and distinctiveness of the relationship. You address this in the discussion, but maybe briefly address it already here. E.g., add the R values for each month. Are the monthly differences due to weather, e.g. soil moisture? Or rather due to the gaps between the**

[Figure]

three installations (but then you would only have it in the leaf water potential). Or due to phenological processes in the trees? You could also show the scatterplots using symbols for the three months and colours for soil moisture values.

Can you include e.g. SMAP VOD over the area (morning and evening overpasses if available) and (briefly) show main differences/similarities to your in situ VOD?

Formulation/spelling:

**71-72: check sentence structure**

**152: they/the, parentheses**

**322: consider turning around "Fig. 5" and "Fig. 6", or use a different wording than "… because the former" - it's a bit confusing to the reader which figure the second half of the sentence refers to**

**451: Fig. 3 and 4 instead of 5 and 6?**

**461: sensitivity**

---

## Author Comment (AC1) · 8 Dec 2020

"L-band vegetation optical depth as an indicator of plant water potential in a temperate deciduous forest stand" by Nataniel Holtzman et al.

**Reply to Reviewer 1**

General comments:
The manuscript presents the use of tower-based radiometer data in L-band to test the link between estimated VOD from one polarization and plant water potential over the red oak forest. The manuscript reports on an experiment carried out during the 2019 growing season in central Massachusetts, United States. The manuscript is well written and contains worthwhile material that brings up the diurnal variation in VOD and its relationship with the diurnal plant water potential cycle. These results are credible and could further the understanding of the crop water dynamics. However, the intensive fieldwork to collect leaf water potential data has been done for 4 days in July which was quite short in time and it was not enough to make a firm conclusion. Although short filed work experiments have been performed, the presented results can be used in future analysis. Therefore, I think this work can be accepted for publication.

**Response:** We thank the reviewer for their thoughtful review.

Specific comments
1- Please add the Precipitation data and dew data in the manuscripts or supplemental information.

**Response:** We thank the reviewer for catching our oversight in not including information on the source of the precipitation data. We have added a sentence at Line 202 that reads: "For this purpose, we used precipitation data from the Fisher Meteorological Station at Harvard Forest, located in an open field approximately 1.3 km from the site of the radiometer." In addition, we have added a plot of precipitation to the bottom panel of Supplemental Figure 2, as shown below.

The dew data comes from leaf wetness sensors we installed in the canopy, as described in lines 143-146: "Five LWS leaf wetness sensors (METER Environment) were installed in the tower at canopy level on July 10. Each sensor recorded a binary reading (wet or dry) every 10 minutes. Hours where the majority of sensor-minutes were wet were considered wet for the purposes of our analysis; all other hours were considered dry."

Our data submission to the NSIDC archive includes CSV files containing the full time series of precipitation and leaf wetness sensor data. This data is now publicly available at https://nsidc.org/data/SV19MA_VOD/versions/1 with a DOI of 10.5067/2PZJDURUJLWF.

[Figure]

Supplemental Figure 2. Time series of VOD, stem xylem dielectric constant at 70 MHz, stem xylem water potential, soil moisture, and precipitation at Harvard Forest.

2- It would be nice if you compare VOD results from SMAP satellite with your data for the study area to illustrate the effect of scale on VOD estimation and especially shows how much VOD estimation would be different between your data and SMAP on the evening overpasses.

**Response**: This is a very good suggestion; thank you. We have added a comparison to a SMAP VOD product and plotted the corresponding data in the revised Supplemental Figure 2 (shown above).

In the methods section at lines 239-244 we added the following:

"Finally, we compared our tower-based single-channel VOD retrievals with VOD retrieved from SMAP satellite data using the multi-temporal dual-channel algorithm (MT-DCA) (Konings et al., 2017). The spatial resolution of this SMAP dataset is 9 km. The SMAP pixel containing the Harvard Forest tower site is masked out in the MT-DCA data, as are the adjacent pixels to the west and south, because of proximity to a water body (the Quabbin Reservoir). Thus, we compared our tower-based VOD to the MT-DCA VOD from the adjacent SMAP pixels to the east and north of the tower site."

In the results section at lines 278-286 we also added the following:

"As illustrated in Supplemental Figure 2, the magnitude of VOD retrieved from the tower-based radiometer using the single-channel algorithm is similar to VOD retrieved from the SMAP satellite over nearby pixels using the MT-DCA. This close match adds to our confidence that our retrieved VOD is in a realistic range for the Harvard Forest site. However, VOD from the tower radiometer shows more detailed temporal dynamics than what is seen from SMAP. For example, between August 7 and August 15 the tower VOD first increases and then decreases, following the changes in stem dielectric. In contrast, SMAP VOD shows little change over that time period, likely due to spatial heterogeneity within the SMAP footprint that does not affect the tower radiometer footprint."

Unfortunately, evening VOD retrievals from the multi-temporal dual-channel algorithm are not available.

3- Although some of the previous studies have indicated that the presence of dew cannot affect the VOD, some studies presented its effect on VOD estimation. It would be nice if you can make a plot and add it in the supplementary in which shows the averaged VOD value at each hour from midnight to early morning for example from 12 to 6 for two conditions of the wet and dry canopy. It will help to better investigate the effect of dew on VOD estimation.

**Response:** We thank the reviewer for the suggestion. The figure below shows VOD as well as stem relative dielectric permittivity, averaged at each hour of the morning (shading represents 1 standard error of the mean). The difference between wet and dry canopy conditions in mean VOD is not significant (t-test, $p > 0.05$) at any hour except for 1 AM and 2 AM, during which VOD is significantly lower when the canopy is wet. However, the wet and dry canopy observations represent distinct sets of days, with different plant hydraulic conditions (i.e. leaf and stem water potential), so they are not necessarily directly comparable. It is impossible to say whether any VOD differences are due to differences in canopy wetness or due to differences in those plant hydraulic conditions.

Indeed, as shown in the lower panel of the figure below, the wet canopy observations are on days with a lower stem dielectric on average, i.e. a smaller internal vegetation water content. Note that leaf water potential is also expected to vary between these wet and dry canopy days, but leaf water potential measurements were not available on most days include in this plot. Overall, the patterns in the hourly VOD analysis cannot be attributed to the presence of canopy wetness alone.

Because the results of this hourly VOD analysis do not provide any additional information about the effects of canopy wetness, we have not included it in the revised manuscript.

[Figure]

4- In figure 3, the temporal variation in VOD is different for each day. It would be nice if you could discuss the reason for the different daily trends. Also, the decreasing trend that you mention from July 11 through July 14 is not visible from the graph and it would be useful if you can fit a line to show that trend in the supplementary material.

**Response:** We have added a discussion of the different daily patterns of VOD at lines 282-285: "There is additional variation on VOD on top of this general diurnal pattern, which is at least partially attributable to transient meteorological conditions. For example, around 1 PM on July 10, the weather at the site changed from sunny to cloudy for an hour, leading to temporarily decreased transpiration rate and thus causing plant water potential and VOD to increase during that hour (Figure 3)."

We thank the reviewer for noticing that the decreasing trend mentioned in the manuscript is not visible. To avoid confusion, we have now deleted it.

5- In figure 7, you fit a line to the scatter plots and consider a linear relationship. By checking data of 3 months, we can see that the data in September shows some linear trend as the fitted line shows but the data of June and July are not fitted to that line. So, I think it would be better for this plot that you will use the Spearman rank correlation method and also shows the correlation for each month separately.

**Response:** We have added a table (shown below) in the supplemental information, with both Pearson and Spearman correlations for each separate month, for each of the three scatter plots in figure 7.

**Stem water potential and stem dielectric**

| Period | R | ρ |
|--------|------|------|
| All | 0.65 | 0.47 |
| July | 0.53 | 0.58 |
| Aug | 0.40 | 0.28 |
| Sept | 0.66 | 0.51 |

**Stem dielectric and VOD**

| Period | R | ρ |
|--------|------|------|
| All | 0.68 | 0.60 |
| July | 0.31 | 0.36 |
| Aug | 0.33 | 0.41 |
| Sept | 0.74 | 0.69 |

**Stem water potential and VOD**

| Period | R | ρ |
|--------|------|------|
| All | 0.65 | 0.54 |
| July | 0.32 | 0.31 |
| Aug | 0.66 | 0.53 |
| Sept | 0.71 | 0.64 |

Supplemental Table 1. Pearson correlations (R) and Spearman rank correlations (ρ) for the three pairs of variables shown as scatter plots in Figure 7, for all data and individually for each of the three periods that the stem psychrometers were installed (corresponding to three months).

We also added the following text to the discussion section 4.1 at line 401:
"Looking at the three installations separately, the highest correlations between all 3 pairs of variables are found in September (Supplemental Table 1). This may be due to dry conditions at that time creating a wider range of stem water potential and stem dielectric values, providing increased signal during September for the same amount of noise."

---

## Author Comment (AC2) · 8 Dec 2020

"L-band vegetation optical depth as an indicator of plant water potential in a temperate deciduous forest stand" by Nataniel Holtzman et al.

**Reply to Reviewer 2**

In this manuscript, the relationship between VOD and plant water potential is analysed using an L-band radiometer and in situ measurements of stem xylem and leaf water potential and dielectric constants. In addition to the relationship in general, the authors investigate diurnal changes of VOD and the sensitivity of VOD to the stem and leaf water potential, respectively. The authors provide a comprehensive overview of basic plant hydraulics, the applied VOD retrieval and all conducted measurements. A weakness of the study is the limited number of in situ samples especially from leaves, but the authors are aware of that and describe the associated uncertainties. Despite the low number of samples, the findings presented in the manuscript are interesting and contribute to a better understand the variables which affect VOD over temperate forest.

**Response:** We thank the reviewer for their thoughtful review.

**266f: In the second half of September, VOD, stem dielectric constant and potential drop significantly. I saw that you refer to this later but consider mentioning it already here.**

**Response:** We thank the reviewer for this suggestion, and added a mention at line 266: "The minimum values of VOD, stem water potential, and soil moisture are all achieved during the same few days in mid-September (Supplemental Figure 2)."

Figure 3c: Do you have an idea why one of the stem water potential curves is very close to leaf water potential on July 10?

**Response:** The stem water potentials were measured on 3 trees within the footprint of the radiometer, while the leaf water potentials were measured on a different set of 5 trees next to the tower but not in the footprint. Although leaf potentials are generally lower than stem potentials on the same tree in daytime due to being more directly affected by transpiration, it is plausible that a leaf potential from one tree may be similar to a stem potential from another tree. We will add a clarifying sentence to the figure caption: "Note that stem water potential and leaf water potential were measured on different sets of trees."

Figure 4: The VOD curve presents the average from April-October. You show in the supplement Fig. 3 that the VOD from July 9-12 does not differ much from the April-October average apart from the absolute values. Have you considered showing both figures in the main part of the manuscript, or adding the July-VOD to Fig. 4? In my opinion this would add information, as April-October is almost the entire growing season, whereas in mid-July not many LAI/biomass-related dynamics occur in temperate forest.

**Response:** We thank the reviewer for the helpful suggestion. Indeed, a combined figure showing the early July VOD diurnal cycle along with the April-October one works well, as shown below. We will include this figure in the main manuscript and delete Supplemental Figure 3.

[Figure]

Figure 4. Average diurnal cycles of VOD and plant water potential. Shaded area is a range of 1 standard error for VOD.

**296f/Figure 5a: The VOD-leaf water potential relationship also seems to break down, but during morning and evening hours and at a leaf potential around -0.5 MPa. Can you elaborate on this?**

**Response:** We agree with the reviewer that there is more noise in the VOD-leaf water potential relationship above -0.5 MPa than below it. On the other hand, it is possible that there would be more noise below that water potential as well if we had measured leaf water potential on more days; it is difficult to make conclusions with the small sample size we have.

Furthermore, the relationship has no statistical change in slope at -0.5 MPa. We verified this lack of a breakpoint by fitting the following model that represents a piecewise linear relationship:
$$VOD = \beta_0 + \beta_1\psi + \beta_2\mathbb{I}[\psi > -0.5] * (\psi - (-0.5))$$
The last term contains an indicator function that is 1 when water potential is greater than -0.5 MPa and 0 otherwise. This term is constructed so the model will still be continuous at -0.5 MPa regardless of a change in slope. The R-squared of this model is 0.58, the same as that of the original simple linear model without a breakpoint; thus adding the breakpoint does not make the model more accurate. The piecewise model also has a less negative Akaike information criterion

value compared to the simple linear model (-45.5 compared to -47.5) which indicates that the simple linear model is to be preferred in a model selection context.

Figure 5/6: You obtain a much better relationship in Fig. 6c than in 5a, just by leaving out the measurements during which you did not measure the leaf dielectric constant (R=89 vs. R=76, no "break down"/vertical linear relationship at -0.5 MPa in Fig. 6c). Can you explain if there is any reason for this? When did you measure the leaf dielectric constant, when not - just randomly?

**Response:** This is a good observation. Leaf permittivity was measured during July 9, 10, and 11, but not measured on July 12 due to the sensor being used for a different experiment that day.

**324f/Figure 7: I agree that there is a linear relationship over the entire growing season. But when looking at the individual months, there is a clear difference in the slope and distinctiveness of the relationship. You address this in the discussion, but maybe briefly address it already here. E.g., add the R values for each month. Are the monthly differences due to weather, e.g. soil moisture? Or rather due to the gaps between the three installations (but then you would only have it in the leaf water potential). Or due to phenological processes in the trees? You could also show the scatterplots using symbols for the three months and colours for soil moisture values.**

**Response:** We have added a table (shown below) in the supplemental information with both Pearson and Spearman correlations for each separate month, for each of the three scatter plots in figure 7.

We have added the following to the discussion section 4.1 at line 401:
> "Looking at the three installations separately, the highest correlations between all 3 pairs of variables are found in September (Supplemental Table 1). This may be due to dry conditions at that time creating a wider range of stem water potential and stem dielectric values, providing increased signal during September for the same amount of noise."

As discussed in the last two paragraphs of section 4.1, we attribute changes from month to month in the form of the VOD-stem dielectric relationship to changes in the individual tree containing the dielectric probe that were not representative of the whole stand. We also attribute changes in slope in the VOD-stem water potential relationship to the trees simply reaching especially low water potentials late in the growing season, at which point the pressure-volume curve (and thus the water potential-VOD curve) may become more non-linear.

**Stem water potential and stem dielectric**

| Period | R | ρ |
|--------|------|------|
| All | 0.65 | 0.47 |
| July | 0.53 | 0.58 |
| Aug | 0.40 | 0.28 |
| Sept | 0.66 | 0.51 |

**Stem dielectric and VOD**

| Period | R | ρ |
|--------|------|------|
| All | 0.68 | 0.60 |
| July | 0.31 | 0.36 |
| Aug | 0.33 | 0.41 |
| Sept | 0.74 | 0.69 |

**Stem water potential and VOD**

| Period | R | ρ |
|--------|------|------|
| All | 0.65 | 0.54 |
| July | 0.32 | 0.31 |
| Aug | 0.66 | 0.53 |
| Sept | 0.71 | 0.64 |

Supplemental Table 1. Pearson correlations (R) and Spearman rank correlations (ρ) for the three pairs of variables shown as scatter plots in Figure 7, for all data and individually for each of the three periods that the stem psychrometers were installed (corresponding to three months).

Can you include e.g. SMAP VOD over the area (morning and evening overpasses if available) and (briefly) show main differences/similarities to your in situ VOD?

**Response**: This is a very good suggestion, which was also made by Reviewer 1. We have added a comparison to a SMAP VOD product, discussed as follows, and plotted the corresponding data in the revised Supplemental Figure 2 (shown below).

In the methods section at lines 239-244 we added the following:
"Finally, we compared our tower-based single-channel VOD retrievals with VOD retrieved from SMAP satellite data using the multi-temporal dual-channel algorithm (MT-DCA) (Konings et al., 2017). The spatial resolution of this SMAP dataset is 9 km. The SMAP pixel containing the Harvard Forest tower site is masked out in the MT-DCA data, as are the adjacent pixels to the west and south, because of proximity to a water body (the Quabbin Reservoir). Thus, we compared our tower-based VOD to the MT-DCA VOD from the adjacent SMAP pixels to the east and north of the tower site."

In the results section at lines 278-286 we added the following:
"As illustrated in Supplemental Figure 2, the magnitude of VOD retrieved from the tower-based radiometer using the single-channel algorithm is similar to VOD retrieved from the

SMAP satellite over nearby pixels using the MT-DCA. This close match adds to our confidence that our retrieved VOD is in a realistic range for the Harvard Forest site. However, VOD from the tower radiometer shows more detailed temporal dynamics than what is seen from SMAP. For example, between August 7 and August 15 the tower VOD first increases and then decreases, following the changes in stem dielectric. In contrast, SMAP VOD shows little change over that time period, likely due to spatial heterogeneity within the SMAP footprint that does not affect the tower radiometer footprint."

Unfortunately, evening VOD retrievals from the multi-temporal dual-channel algorithm are not available.

[Figure]

Supplemental Figure 2. Time series of VOD, stem xylem dielectric constant at 70 MHz, stem xylem water potential, soil moisture, and precipitation at Harvard Forest.

**We also thank the reviewer for catching several typographical errors in the manuscript. These errors have now been fixed as follows:**

**71-72: check sentence structure**
**Line 71 now reads** "Momen et al. (2017) compared fluctuations in satellite-based X-band VOD to *in situ* leaf water potential measurements in three forest and woodland sites"

**Line 152 now reads** "Depths are approximate, as the sensing volume varies depending on soil moisture status and signal magnitude: it is strongest close to the sensor and decreases away from a sensor."

**Line 322 now reads** "The VOD-leaf potential correlation in Fig. 6 is different from that in Fig. 5, because Fig.6 is limited to leaf water potential observations that coincided with a leaf dielectric constant observation."

**Line 451 now reads** "Between 5 AM and 8 AM, stem xylem potential stays high while VOD and leaf potential begin to decrease, as shown in Figs. 3 and 4."

Line 461 now reads "Nevertheless, the relatively larger sensitivity of VOD to leaf water potential than to stem water potential is notable because most of a tree's mass is in its trunk."